# Online Bayesian Moment Matching for Topic Modeling with Unknown Number of Topics

**Wei-Shou Hsu and Pascal Poupart**
David R. Cheriton School of Computer Science
University of Waterloo
Wateroo, ON N2L 3G1
{wwhsu,ppoupart}@uwaterloo.ca

## Abstract

Latent Dirichlet Allocation (LDA) is a very popular model for topic modeling as well as many other problems with latent groups. It is both simple and effective. When the number of topics (or latent groups) is unknown, the Hierarchical Dirichlet Process (HDP) provides an elegant non-parametric extension; however, it is a complex model and it is difficult to incorporate prior knowledge since the distribution over topics is implicit. We propose two new models that extend LDA in a simple and intuitive fashion by directly expressing a distribution over the number of topics. We also propose a new online Bayesian moment matching technique to learn the parameters and the number of topics of those models based on streaming data. The approach achieves higher log-likelihood than batch and online HDP with fixed hyperparameters on several corpora. The code is publicly available at https://github.com/whsu/bmm.

## 1 Introduction

Latent Dirichlet Allocation (LDA) [3] recently emerged as the dominant framework for topic modeling as well as many other applications with latent groups. The Hierarchical Dirichlet Process (HDP) [18] provides an elegant extension to LDA when the number of topics (latent groups) is unknown. The non-parametric nature of HDPs is quite attractive since HDPs effectively allow an unbounded number of topics to be inferred from the data. There is also a rich mathematical theory underlying HDPs as well as attractive metaphors (e.g., stick breaking process, Chinese restaurant franchise) to ease the understanding by those less comfortable with non-parametric statistics [18]. That being said, HDPs are not perfect. They do not expose an explicit distribution over the topics that could allow practitioners to incorporate prior knowledge and to inspect the model's posterior confidence in different number of topics. Furthermore, the implicit distribution over the number of topics is restricted to a regime where the number of topics grows logarithmically with the amount of data in expectation [18]. For instance, this growth rate is insufficient for applications that exhibit a power law distribution [6] – a generalization of the HDP known as the hierarchical Pitman-Yor process [21] is often used instead. Existing inference algorithms for HDPs (e.g., Gibbs sampling [18], variational inference [19, 24, 23, 4, 17]) are also fairly complex. As a result, practitioners often stick with LDA and estimate the number of topics by repeatedly evaluating different number of topics by cross-validation; however, this is an expensive procedure.

We propose two new models that extend LDA in a simple and intuitive fashion by directly expressing a distribution over the number of topics under the assumption that an upper bound on the number of topics is available. When the amount of data is finite, this assumption is perfectly fine since there cannot be more topics than the amount of data. Otherwise, domain experts can often define a suitable range for the number of topics and if they plan to inspect the resulting topics, they cannot inspect

an unbounded number of topics. We also propose a novel Bayesian moment matching algorithm to compute a posterior distribution over the model parameters and the number of topics. Bayesian learning naturally lends itself to online learning for streaming data since the posterior is updated sequentially after each data point and there is no need to go over the data more than once. The main issue is that the posterior becomes intractable. We approximate the posterior after each observed word by a tractable distribution that matches some moments of the exact posterior (hence the name Bayesian Moment Matching). The approach compares favorably to online HDP on several topic modeling tasks.

## 2 Related work

Setting the number of topics to use can be treated as a model selection problem. One solution is to train a topic model multiple times, each time with a different number of topics, and choose the number of topics that minimizes some cost function on a heldout test set. More recently nonparametric Bayesian methods have been used to bypass the model selection problem. Hierarchical Dirichlet process (HDP) [18] is the natural extension of LDA in this direction. With HDP, the number of topics is learned from data as part of the inference procedure. Gibbs sampling [7, 15] and Variational Bayes [3, 20] are by far the most popular inference techniques for LDA. They have been extended to HDP [18, 19, 17]. With the rise of streaming data, online variants of Variational Bayes have also been developed for LDA [8] and HDP [24, 23, 4]. The first online variational technique [24] used a truncation that effectively bounds the number of topics while subsequent techniques [23, 4] avoid any fixed truncation to fully exploit the non-parametric nature of HDP. These online variational techniques perform stochastic gradient ascent on mini-batches, which reduces their data efficiency, but improves computational efficiency.

We propose two new models that are simpler than HDP and express a distribution directly on the number of topics. We extend online Bayesian moment matching (originally designed for LDA with a fixed number of topics [14]) to learn the number of topics. This technique avoids mini-batches. It approximates Bayesian learning by Assumed Density Filtering [13], which can be thought as a single forward iteration of Expectation Propagation [12]. Note that Bayesian moment matching is different from frequentist moment matching techniques such as spectral learning [1, 2, 9, 11]. In BMM, we compute a posterior over the parameters of the model and approximate the posterior with a simpler distribution that matches some moments of the exact posterior. In spectral learning, moments of the empirical distribution of the data are used to find parameters that yield the same moments in the model. This is usually achieved by a spectral (or tensor) decomposition of the empirical moments, hence the name spectral learning. Although both BMM and spectral learning use the method of moments, they match different moments in different distributions resulting in completely different algorithms. While stochastic gradient descent can be used to compute tensor decompositions in an online fashion [5, 10], no online variant of spectral learning has been developed to infer the number of topics in LDA.

## 3 Models

We investigate the problem of online clustering of grouped discrete observations. Using terminology from text processing, we will call each observation a word and each group a document. The observed data set is then a corpus of $N$ words, $\{w_n\}_{n=1}^N$, along with the IDs, $\{d_n\}_{n=1}^N$, of the documents to which these words belong. We will let $D$ denote the number of documents and $V$ the number of distinct words in the vocabulary. Figure 1 shows the generative models we are considering.

The basic model is LDA, in which the number of the topics $T$ is fixed. We propose two extensions to the basic model where the parameter $T$ is unknown and inferred from data, with the assumption that $T$ ranges from 1 to $K$. Each $\vec{\theta}$ specifies the topic distribution of a document, while each $\vec{\phi}$ specifies the word distribution of a topic. In the rest of the paper, we will use $\Theta$ to denote the collection of all $\vec{\theta}$'s and $\Phi$ the collection of all $\vec{\phi}$'s in the model.

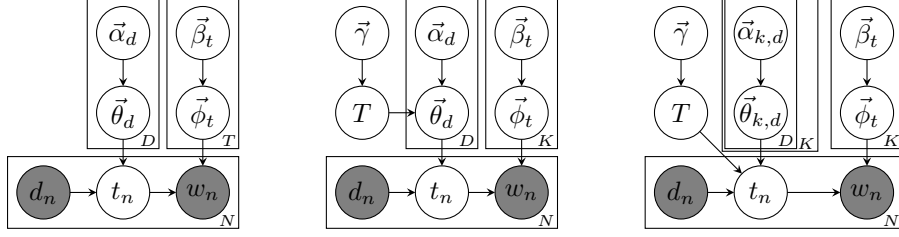

Figure 1: Graphical representations of basic model with fixed number of topics (left), degenerate Dirichlet model (middle), and triangular Dirichlet model (right)

## 3.1 Degenerate Dirichlet model

The generative process of the degenerate Dirichlet model (DDM), as shown in the middle in Figure 1, works by first sampling the hyperparameters $\vec{\gamma}$, $\{\vec{\alpha}_d\}_{d=1}^D$, and $\{\vec{\beta}_t\}_{t=1}^K$. The parameters $T$, $\{\vec{\theta}_d\}_{d=1}^D$, and $\{\vec{\phi}_t\}_{t=1}^K$ are then sampled from the following conditional distributions:

$$P(T|\vec{\gamma}) = \text{Discrete}(T; \vec{\gamma})$$

$$P(\vec{\theta}_d|\vec{\alpha}_d, T) = \text{Dir}(\vec{\theta}_d; \vec{\alpha}_d, T)$$

$$P(\vec{\phi}_t|\vec{\beta}_t) = \text{Dir}(\vec{\phi}_t; \vec{\beta}_t)$$

where $\text{Dir}(\vec{\theta}_d; \vec{\alpha}_d, T)$ denotes a degenerate Dirichlet distribution $\text{Dir}(\vec{\theta}_d; \vec{\alpha}'_d)$ with

$$\alpha'_{d,t} = \begin{cases} \alpha_{d,t} & \text{for } t \leq T \\ 0 & \text{for } t > T \end{cases}$$

and $\text{Discrete}(T; \vec{\gamma})$ is the general discrete distribution with probability $P(T = k) = \gamma_k$ for $k = 1, \ldots, K$. Finally, the $N$ observations are generated by first sampling the topic indicators $t_n$ according to the distribution $P(t_n|d_n, \Theta) = \theta_{d_n, t_n}$. Note that since $\vec{\theta}_{d_n}$ is sampled from a degenerate Dirichlet, we have $\theta_{d_n, t_n} = 0$ for $t_n > T$. Given $t_n$, the words are then sampled according to the categorical distribution $P(w_n|t_n, \Phi) = \phi_{t_n, w_n}$.

## 3.2 Triangular Dirichlet model

The triangular Dirichlet model (TDM), shown on the right in Figure 1, works in a similar way except the document-topic distribution $\Theta$ is represented by a three-dimensional array that is also indexed by the number of topics $T$ in addition to the document ID $d$ and the topic ID $t$. Given $T$ and $d$, the topic $t$ is drawn according to the probability $P(t|d, \Theta, T) = \theta_{T,d,t}$ for $1 \leq t \leq T$. The array $\Theta$ therefore has a triangular shape in the first and third dimension. Again, we place a Dirichlet prior on each $\vec{\theta}_{k,d}$: $P(\vec{\theta}_{k,d}|\vec{\alpha}_{k,d}) = \text{Dir}(\vec{\theta}_{k,d}; \vec{\alpha}_{k,d})$. In this case, however, $\vec{\theta}_{k,d}$ has no dependence on $T$.

# 4 Bayesian update by moment matching

Let $P_n(\Theta, \Phi, T)$ denote the joint posterior probability of $\Theta$, $\Phi$, and $T$ after seeing the first $n$ observations. Then[1]

$$P_n(\Theta, \Phi, T) = P(\Theta, \Phi, T|w_{1:n})$$

$$= \frac{1}{c_n} \sum_{t_n=1}^K P(t_n|\Theta, T) P(w_n|\Phi, t_n) P_{n-1}(\Theta, \Phi, T) \tag{1}$$

where $c_n = P(w_n|w_{1:n-1})$.

From (1) we can see that after seeing each new observation $w_n$, the number of terms in the posterior is increased by a factor of $K$, resulting in an exponential complexity for exact Bayesian update. Therefore, we will instead approximate $P_n$ by a different distribution, whose parameters will be estimated by moment matching.

## 4.1 Approximating distribution

To make the inference tractable, we approximate $P_n$ using a factorized distribution: $P_n(\Theta, \Phi, T) = f_\Theta(\Theta) f_\Phi(\Phi) f_T(T)$.

For TDM, we choose the factorized distribution to have the exact same form as the prior distribution, i.e.,

$$f_\Theta(\Theta) = \prod_{k=1}^{K} \prod_{d=1}^{D} \text{Dir}(\vec{\theta}_{k,d}; \vec{\alpha}_{k,d}) \tag{2}$$

$$f_\Phi(\Phi) = \prod_{t=1}^{K} \text{Dir}(\vec{\phi}_t; \vec{\beta}_t) \tag{3}$$

$$f_T(T) = \text{Discrete}(T; \vec{\gamma}) \tag{4}$$

For DDM, we use the same $f_\Phi$ and $f_T$, but rather than choosing $f_\Theta$ as degenerate Dirichlets again, we instead approximate the posterior over $\Theta$ using proper Dirichlet distributions to decouple $\Theta$ from $T$:

$$f_\Theta(\Theta) = \prod_{d=1}^{D} \text{Dir}(\vec{\theta}_d; \vec{\alpha}_d) \tag{5}$$

## 4.2 Moment matching

Let $x$ be a random variable with distribution $p(x)$. The $i$-th moment of $x$ about zero is defined as the expectation of $x^i$ over $p$, and we denote it by $M_{x^i}(p)$:

$$M_{x^i}(p) = E_p\left[x^i\right] \tag{6}$$

For a $K$-dimensional Dirichlet distribution $\text{Dir}(x_1, \ldots, x_K; \tau_1, \ldots, \tau_K)$, we can uniquely solve for the parameters $\tau_1, \ldots, \tau_K$ if we have $K-1$ first moments, $M_{x_1}, \ldots, M_{x_{K-1}}$, and one second moment, $M_{x_1^2}$. Given the moments, we can determine the Dirichlet parameters as

$$\tau_k = M_{x_k} \frac{M_{x_1} - M_{x_1^2}}{M_{x_1^2} - M_{x_1}^2} \tag{7}$$

for $k = 1, \ldots, K$. Therefore, we can compute the parameters for $f_\Theta$ and $f_\Phi$ using (7): for $\alpha_d$, replace $\tau_k$ with $\alpha_{d,k}$ and $x_k$ with $\theta_{d,k}$; and for $\beta_t$, replace $\tau_k$ with $\beta_{t,k}$ and $x_k$ with $\phi_{t,k}$.

The parameters for $\text{Discrete}(T; \vec{\gamma})$ are estimated directly as

$$\gamma_k = E[\delta_{T,k}] \tag{8}$$

where $\delta$ denotes the the Kronecker delta

$$\delta_{i,j} = \begin{cases} 1 & \text{if } i = j \\ 0 & \text{if } i \neq j \end{cases}. \tag{9}$$

## 4.3 Moment computation

From (7) and (8), we see that to approximate $P_n$ by moment matching, we need to compute the first and second moments of $\Theta$ and $\Phi$ as well as the expectation $E[\delta_{T,k}]$ with respect to $P_n$. They can be calculated using the Bayesian update equation (1).

To keep the notation uncluttered, let $S_{\vec{x},:m}$ denote the sum of the first $m$ elements in a vector $\vec{x}$ and $S_{\vec{x}}$ the sum of all elements in $\vec{x}$. We can then compute the moments of DDM as follows:

$$c_n = \sum_{T=1}^{K} \gamma_T \sum_{t_n=1}^{T} \frac{\alpha_{d_n,t_n}}{S_{\vec{\alpha}_{d_n},:T}} \frac{\beta_{t_n,w_n}}{S_{\vec{\beta}_{t_n}}} \tag{10}$$

$$E_{P_n}[\delta_{T,k}] = \frac{1}{c_n} \gamma_k \sum_{t_n=1}^{k} \frac{\alpha_{d_n,t_n}}{S_{\vec{\alpha}_{d_n},:k}} \frac{\beta_{t_n,w_n}}{S_{\vec{\beta}_{t_n}}} \tag{11}$$

$$M_{\theta_{d,t}}(P_n) = \frac{1}{c_n} \sum_{T=t}^{K} \gamma_T \sum_{t_n=1}^{T} \frac{\alpha_{d_n,t_n}}{S_{\vec{\alpha}_{d_n},:T}} \frac{\beta_{t_n,w_n}}{S_{\vec{\beta}_{t_n}}} \frac{\alpha_{d,t} + \delta_{d,d_n}\delta_{t,t_n}}{S_{\vec{\alpha}_d,:T} + \delta_{d,d_n}} \tag{12}$$

$$M_{\theta_{d,t}^2}(P_n) = \frac{1}{c_n} \sum_{T=t}^{K} \gamma_T \sum_{t_n=1}^{T} \frac{\alpha_{d_n,t_n}}{S_{\vec{\alpha}_{d_n},:T}} \frac{\beta_{t_n,w_n}}{S_{\vec{\beta}_{t_n}}} \frac{\alpha_{d,t} + \delta_{d,d_n}\delta_{t,t_n}}{S_{\vec{\alpha}_d,:T} + \delta_{d,d_n}} \frac{\alpha_{d,t} + 1 + \delta_{d,d_n}\delta_{t,t_n}}{S_{\vec{\alpha}_d,:T} + 1 + \delta_{d,d_n}} \tag{13}$$

$$M_{\phi_{t,w}}(P_n) = \frac{1}{c_n} \sum_{T=1}^{K} \gamma_T \sum_{t_n=1}^{T} \frac{\alpha_{d_n,t_n}}{S_{\vec{\alpha}_{d_n},:T}} \frac{\beta_{t_n,w_n}}{S_{\vec{\beta}_{t_n}}} \frac{\beta_{t,w} + \delta_{t,t_n}\delta_{w,w_n}}{S_{\vec{\beta}_t} + \delta_{t,t_n}} \tag{14}$$

$$M_{\phi_{t,w}^2}(P_n) = \frac{1}{c_n} \sum_{T=1}^{K} \gamma_T \sum_{t_n=1}^{T} \frac{\alpha_{d_n,t_n}}{S_{\vec{\alpha}_{d_n},:T}} \frac{\beta_{t_n,w_n}}{S_{\vec{\beta}_{t_n}}} \frac{\beta_{t,w} + \delta_{t,t_n}\delta_{w,w_n}}{S_{\vec{\beta}_t} + \delta_{t,t_n}} \frac{\beta_{t,w} + 1 + \delta_{t,t_n}\delta_{w,w_n}}{S_{\vec{\beta}_t} + 1 + \delta_{t,t_n}} \tag{15}$$

For TDM, the moments are computed similarly except that $T$ is used to index into $\alpha$ rather than to take partial sums. The equations are included in the supplement.

### 4.4 Parameter update

For TDM, the approximating distribution for the posterior has the exact same form as the prior; therefore, the parameters we compute for $P_n$ in the $n$-th update can be used directly as the parameters for the prior in the $(n+1)$-th update.

However, for DDM, the prior for $\Theta$ consists of degenerate Dirichlet distributions conditionally dependent on $T$, whereas the approximating distribution for the posterior is a fully factorized distribution with proper Dirichlets. Therefore, we have to make a further approximation to match the parameters of the two distributions.

When $P_n$ is being used as the prior in the $(n+1)$-th update, we use the same $\alpha$ that was obtained by moment matching during the $n$-th update, but it now has a different meaning. During the $n$-th update, $\alpha$ is computed as parameters of proper Dirichlet distributions, but in the next update, it is used as parameters of a weighted sum of degenerate Dirichlet distributions. As a result, the DDM has a natural bias towards smaller number of topics.

### 4.5 Algorithm summary

In summary, starting from a prior distribution, the algorithm successively updates the posterior by first computing the exact moments according to the Bayesian update equation (1), and then updating the parameters by matching the moments with those of an approximating distribution. In the case of TDM, the approximating distribution has the same form as the prior, whereas a simplified distribution is used for DDM. Algorithm 1 summarizes the procedure for the two models.

---

**Algorithm 1** Online Bayesian moment matching algorithm

---

1: Initialize $\alpha$, $\beta$, and $\vec{\gamma}$.
2: **for** $n = 1, \ldots, N$ **do**
3:    Read the $n$-th observation $(d_n, w_n)$.
4:    Compute moments according to (10)–(15) for DDM or equations in supplement for TDM.
5:    Update $\alpha$, $\beta$, and $\vec{\gamma}$ according to (7) and (8) with appropriate substitutions.
6: **end for**

---

## 5 Experiments

In this section, we discuss our experiments on a synthetic dataset and three real text corpora. The TDM and DDM implementations are available at `https://github.com/whsu/bmm`. For both models we initialized the hyperparameters to be $\alpha_{d,t} = 1$ and $\beta_{t,w} = \frac{1}{\sqrt{V}}$ for all $d$, $t$, and $w$. The reason that $\beta_{t,w}$ was not initialized to 1 was to encourage the algorithm to find topics with more concentrated word distributions.

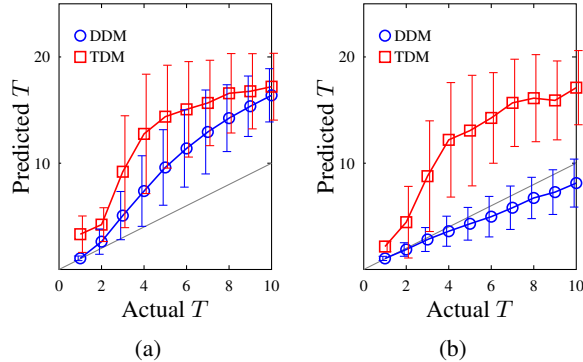

Figure 2: Number of topics discovered by the DDM and TDM on synthetic datasets using (a) uniform prior and (b) exponentially decreasing prior on $T$. The results are averaged over 100 randomly generated datasets for each actual $T$. Error bars show plus/minus one standard deviation. Gray line indicates the true number of topics that generated the datasets.

## 5.1 Synthetic data

We first ran some tests on synthetic data to see how well the models estimate the number of topics. For this experiment, the actual number of topics $T$ was varied from 1 to 10, and for each value of $T$, we generated 100 random datasets with $D = 100$, $V = 200$, and $N =$100,000. Each random dataset was created by first sampling $\Theta$ from $\text{Dir}(\vec{\alpha}_d|0.05)$ and $\Phi$ from $\text{Dir}(\vec{\beta}_t|0.1)$. The observations were then sampled from $\Theta$ and $\Phi$.

We set $K = 20$ and used the uniform prior $P(T) = \frac{1}{K}$ for $T = 1, \ldots, K$. The estimated number of topics is shown in Figure 2(a). Both models were able to discover more topics as the actual number of topics increases. They tend to overestimate the number of topics because the initial value $\beta_{t,w} = \frac{1}{\sqrt{V}}$ encourages topics with smaller number of words.

However, in both models, the modeler has direct control over the number of topics. If there is reason to believe the data come from a smaller number of topics, the modeler can change the prior distribution on $T$ accordingly as is typical in a Bayesian framework.

For this example, we also tested on an exponentially decreasing prior $P(T) \propto e^{-T}$ for $T = 1, \ldots, K$. The results are shown in Figure 2(b). In this case, TDM shows a slight decrease than with a uniform prior, whereas DDM produces an estimate that is close to the true number of topics.

## 5.2 Text modeling

We compare the two proposed models by using them to model the distributions of three real text corpora containing Reuters news articles, NIPS conference proceedings, and Yelp reviews. We also include online HDP (oHDP) in the comparisons, as well as the basic moment matching (basic MM) algorithm with different values of $T$. For online HDP, we used the gensim 0.10.3 [16] implementation with the default parameters except for the top-level truncation, which we set equal to the maximum number of topics we used for DDM and TDM. Because DDM and TDM do not estimate a global alpha as oHDP, for oHDP we include the results with both uniform alpha (oHDP unif) and alpha that is learned (oHDP alpha).

We followed a similar experimental setup as in [22, 4]. Each dataset was divided into a training set $\mathcal{D}_{\text{train}}$ and a test set $\mathcal{D}_{\text{test}}$ based on document IDs. The words in the test set were further split into two subsets $\mathcal{W}_1$ and $\mathcal{W}_2$, where $\mathcal{W}_1$ contains the words in the first half of each document in the test set, and $\mathcal{W}_2$ contains the second half. The evaluation metric used is the per-word log likelihood $L = \frac{\log p(\mathcal{W}_2|\mathcal{W}_1, \mathcal{D}_{\text{train}})}{|\mathcal{W}_2|}$ where $|\mathcal{W}_2|$ denotes the total number of tokens in $\mathcal{W}_2$.

For each experiment we also report the number of topics inferred by DDM, TDM. We do not report this number for online HDP because it is not returned by the implementation.

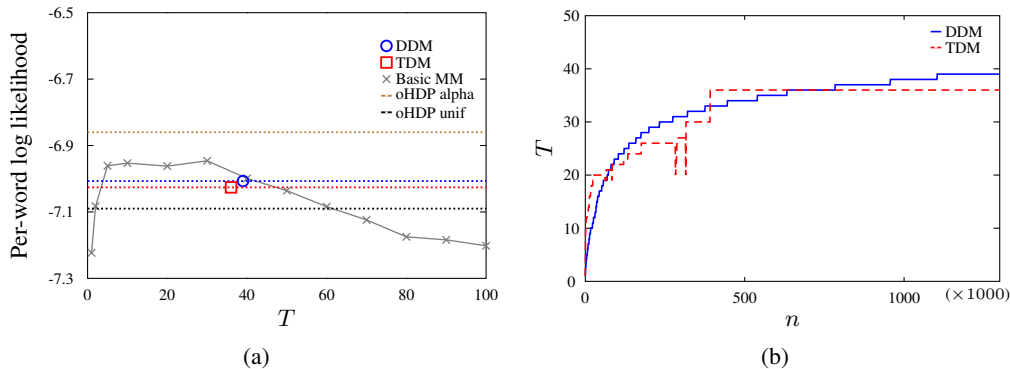

Figure 3: Text modeling on Reuters-21578: (a) Per-word test log likelihood and (b) Number of topics found as a function of number of observations.

### 5.2.1 Reuters-21578

The Reuters-21578 corpus contains 21,578 Reuters news articles in 1987. For this dataset, we divided the data into training and test sets according to the `LEWISSPLIT` attribute that is available as part of the distribution at `http://www.daviddlewis.com/resources/testcollections/reuters21578/`. The text was passed through a stemmer, and stopwords and words appearing in five or fewer documents were removed. This resulted in a total of 1,307,468 tokens and a vocabulary of 7,720 distinct words. We chose $K$ to be 100 for both models with uniform prior $P(T) = \frac{1}{K}$. Figure 3(a) shows the experimental results.

DDM discovered 39 topics while TDM found 36, and they both achieved similar per-word log likelihood as the best models with fixed $T$ showing that they were able to automatically determine the number of topics necessary to model the data.

While both models found the a similar number of topics in the end, they progressed to the final values in different ways. Fig. 3(b) shows the number of topics found by the two models as a function of number of observations. DDM shows a logarithmically increasing trend as more words are observed, whereas TDM follows a more irregular progression.

### 5.2.2 NIPS

We also tested the two models on 2,742 articles from the NIPS conference for the years 1988–2004. We used the raw text versions available at `http://cs.nyu.edu/~roweis/data.html` (1988–1999) and `http://ai.stanford.edu/~gal/data.html` (2000–2004). The first set was used as the training set and the second as the test set. The corpus was again passed through a stemmer, and stopwords and words appearing no more than 50 times were removed. After preprocessing we are left with 2,207,106 total words and a vocabulary of 4,383 unique words.

For this dataset we used $K = 400$ with the exponentially decreasing prior. DDM discovered 54 topics, and TDM found 89 topics. Figure 4(a) shows the per-word log likelihood on the test set.

In this experiment, both DDM and TDM obtained closed to the optimal likelihood compared to basic MM.

### 5.2.3 Yelp

In our third experiment, we tested the models on a subset of the Yelp Academic Dataset (`http://www.yelp.com/dataset_challenge`). We took the 129,524 reviews in the dataset that were given to businesses in the Food category. The reviews were randomly split so that 70% were used for training and 30% for testing.

Similar preprocessing was performed. The corpus was passed through a stemmer, and stopwords and words appearing no more than 50 times were removed. After preprocessing the corpus contains a total of 5,317,041 words and a vocabulary of 5,640 distinct words.

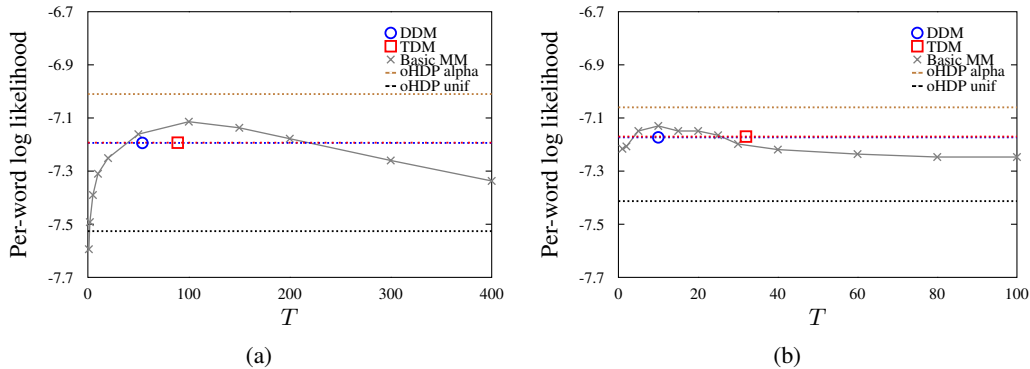

Figure 4: Per-word test log likelihood of (a) NIPS and (b) Yelp.

For this dataset, we tested with $K$=100 using the exponentially decreasing prior on $T$. Figure 4(b) shows the per-word log likelihood on the test set. DDM found the optimal number of topics while both models achieved close to best likelihood on the test set compared to basic MM.

### 5.2.4 Comparison with online HDP

Because DDM and TDM do not estimate the global alpha, in the experiments we compute the test likelihood using a uniform alpha. If we also use a uniform alpha for online HDP, DDM and TDM achieve higher test likelihood. However, online HDP is able to learn the global alpha, which results in higher likelihood. This is a shortcoming of our models, and we are exploring ways to estimate the global alpha.

### 5.3 Additional experimental results

Additional experimental results may be found in the supplement, including running time of the experiments and samples of topics discovered in the Reuters and NIPS corpora, as well as experiments on using the models as dimensionality reduction preprocessors in text classification.

## 6 Conclusions

In this paper we proposed two topic models that can be used when the number of topics is not known. Unlike nonparametric Bayesian models, the proposed models provide explicit control over the prior for the number of topics. We then presented an online learning algorithm based on Bayesian moment matching, and experiments showed that reasonable topics could be recovered using the proposed models. Additional experiments on text classification and visual inspection of the inferred topics show that the clusters discovered were indeed semantically meaningful.

One unsolved problem is that the proposed models do not estimate the global alpha, resulting in lower test likelihood compared to online HDP, which is able to estimate alpha. Developing a robust way to estimate alpha will be the next step to improve the models.

## Footnotes

[1]In the derivations that follow, the dependence on the document IDs $\{d_n\}_{n=1}^D$ and the hyperparameters $\vec{\gamma}$, $\boldsymbol{\alpha}$, and $\boldsymbol{\beta}$ is implicit and not shown.

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
