[Supplementary Material]

# Supplement to Online Bayesian Moment Matching for Topic Modeling with Unknown Number of Topics

**Wei-Shou Hsu and Pascal Poupart**
David R. Cheriton School of Computer Science
University of Waterloo
Wateroo, ON N2L 3G1
{wwhsu,ppoupart}@uwaterloo.ca

## 1  Moment computation for TDM

For TDM, the moments are computed similarly as DDM except that $T$ is used to index into $\boldsymbol{\alpha}$ rather than to take partial sums:

$$c_n = \sum_{T=1}^{K} \gamma_T \sum_{t_n=1}^{T} \frac{\alpha_{T,d_n,t_n}}{S_{\vec{\alpha}_{T,d_n}}} \frac{\beta_{t_n,w_n}}{S_{\vec{\beta}_{t_n}}} \tag{1}$$

$$E_{P_n}\left[\delta_{T,k}\right] = \frac{1}{c_n} \gamma_k \sum_{t_n=1}^{k} \frac{\alpha_{k,d_n,t_n}}{S_{\vec{\alpha}_{k,d_n}}} \frac{\beta_{t_n,w_n}}{S_{\vec{\beta}_{t_n}}} \tag{2}$$

$$M_{\theta_{k,d,t}}\left(P_n\right) = \frac{1}{c_n} \sum_{T=1}^{K} \gamma_T \sum_{t_n=1}^{T} \frac{\alpha_{k,d_n,t_n}}{S_{\vec{\alpha}_{T,d_n}}} \frac{\beta_{t_n,w_n}}{S_{\vec{\beta}_{t_n}}} \frac{\alpha_{k,d,t} + \delta_{k,T}\delta_{d,d_n}\delta_{t,t_n}}{S_{\vec{\alpha}_{T,d}} + \delta_{k,T}\delta_{d,d_n}} \tag{3}$$

$$M_{\theta_{k,d,t}^2}\left(P_n\right) = \frac{1}{c_n} \sum_{T=1}^{K} \gamma_T \sum_{t_n=1}^{T} \frac{\alpha_{k,d_n,t_n}}{S_{\vec{\alpha}_{T,d_n}}} \frac{\beta_{t_n,w_n}}{S_{\vec{\beta}_{t_n}}} \tag{4}$$
$$\frac{\alpha_{k,d,t} + \delta_{k,T}\delta_{d,d_n}\delta_{t,t_n}}{S_{\vec{\alpha}_{T,d}} + \delta_{k,T}\delta_{d,d_n}} \frac{\alpha_{k,d,t} + 1 + \delta_{k,T}\delta_{d,d_n}\delta_{t,t_n}}{S_{\vec{\alpha}_{T,d}} + 1 + \delta_{k,T}\delta_{d,d_n}}$$

$$M_{\phi_{t,w}}\left(P_n\right) = \frac{1}{c_n} \sum_{T=1}^{K} \gamma_T \sum_{t_n=1}^{T} \frac{\alpha_{T,d_n,t_n}}{S_{\vec{\alpha}_{T,d_n}}} \frac{\beta_{t_n,w_n}}{S_{\vec{\beta}_{t_n}}} \frac{\beta_{t,w} + \delta_{t,t_n}\delta_{w,w_n}}{S_{\vec{\beta}_t} + \delta_{t,t_n}} \tag{5}$$

$$M_{\phi_{t,w}^2}\left(P_n\right) = \frac{1}{c_n} \sum_{T=1}^{K} \gamma_T \sum_{t_n=1}^{T} \frac{\alpha_{T,d_n,t_n}}{S_{\vec{\alpha}_{T,d_n}}} \frac{\beta_{t_n,w_n}}{S_{\vec{\beta}_{t_n}}} \frac{\beta_{t,w} + \delta_{t,t_n}\delta_{w,w_n}}{S_{\vec{\beta}_t} + \delta_{t,t_n}} \frac{\beta_{t,w} + 1 + \delta_{t,t_n}\delta_{w,w_n}}{S_{\vec{\beta}_t} + 1 + \delta_{t,t_n}} \tag{6}$$

## 2  Computational complexity

### 2.1  Computational complexity of DDM

First, note that $M_{\theta_{d,t}}$ and $M_{\theta_{d,t}^2}$ do not change if $d \neq d_n$; therefore, we need only update $M_{\theta_{d_n,t}}$ and $M_{\theta_{d_n,t}^2}$ for the $n$th observation, and computing $\boldsymbol{\alpha}$ and the moments of $\Theta$ takes $O(K)$ time for each observation. The bulk of computation is thus on $\boldsymbol{\beta}$ and the moments of $\Phi$.

There are $KV$ parameters in $\boldsymbol{\beta}$. Updating all of them therefore takes $O(KV)$ time. On the other hand, updating the moments of $\Phi$ naively using the equation

$$M_{\phi_{t,w}}(P_n) = \frac{1}{c_n} \sum_{T=1}^{K} \gamma_T \sum_{t_n=1}^{T} \frac{\alpha_{d_n,t_n}}{S_{\vec{\alpha}_{d_n},:T}} \frac{\beta_{t_n,w_n}}{S_{\vec{\beta}_{t_n}}} \frac{\beta_{t,w} + \delta_{t,t_n}\delta_{w,w_n}}{S_{\vec{\beta}_t} + \delta_{t,t_n}} \tag{7}$$

for all $t$ and $w$ has $O(K^3 V)$ complexity. However, (7) can be simplified. First define the auxiliary functions

$$f(T) = \frac{\gamma_T}{S_{\vec{\alpha}_{d_n},:T}}$$

$$g(t_n) = \frac{\alpha_{d_n,t_n}\beta_{t_n,w_n}}{S_{\vec{\beta}_{t_n}}}.$$

Then (7) becomes

$$M_{\phi_{t,w}}(P_n) = \frac{1}{c_n}\left\{ \beta_{t,w} \left[ \sum_{T=1}^{K} f(T) \sum_{t_n=1}^{T} \frac{g(t_n)}{S_{\vec{\beta}_t} + \delta_{t,t_n}} \right] + \frac{\delta_{w,w_n} g(t)}{S_{\vec{\beta}_t} + 1} \left[ \sum_{T=t}^{K} f(T) \right] \right\}$$

The double sum in the square brackets can be precomputed in $O(K^2)$ time for all $t$ by noting that the denominator $S_{\vec{\beta}_t} + \delta_{t,t_n}$ is constant for all $t_n$ except at one value. The complexity to compute $M_{\phi_{t,w}}$ for each observation then becomes $O(KV + K^2)$.

Furthermore, one more simplification is possible by noting that we are making one update for each observed word. Therefore, each update can in fact be made constant time with respect to the vocabulary size $V$. For each update, the same factor is applied to the moments $M_{\phi_{t,w}}$ for all $w \neq w_n$, and we only have to keep track of this common factor instead of making the update for each individual $w$. With these simplifications, this algorithm has $O(K^2)$ complexity per observation and thus runs in $O(NK^2)$ times for a dataset with $N$ observations.

## 2.2 Computational complexity of TDM

TDM has extra $\boldsymbol{\alpha}$ parameters to compute in each iteration. For each observation, we need to update $\alpha_{k,d_n,t}$ for $k = 1, \ldots, K$ and $t = 1, \ldots, K$. Computing $\boldsymbol{\alpha}$ and the moments of $\Theta$ thus takes $O(K^2)$ times. Updating $\boldsymbol{\beta}$ and the moments of $\Phi$ has $O(K^2)$ complexity as well using similar simplifications as for DDM. Computational complexity for each observation is therefore again $O(K^2)$.

## 3 Topics discovered in the Reuters and NIPS corpora

Tables 1 and 2 show the top 20 words from some of the topics discovered by DDM in the Reuters and NIPS corpora. Each topic is given a human-annotated label. As can be seen, the algorithm is able to find reasonable clusters containing semantically related words.

## 4 Running time of text modeling experiments

Table 3 shows how long each algorithm took in the experiments. Here we also include the running time of batch HDP using the implementation available at `http://www.cs.cmu.edu/~chongw/software/hdp.tar.gz`. For basic moment matching, we report the total running times of every $T$ tested (indicated by the crosses in the graphs), and we indicate in parenthesis the running time with the largest value of $T$ tested (400 for NIPS and 100 for Reuters and Yelp).

## 5 Experiment on text classification

In the second experiments with the Reuters-21578 and Yelp datasets, we used the $\Theta$ parameters as the feature vectors in a supervised learning setting. For the Reuters set, we tested the classification performance on the six topics with the most positive examples in the corpus. The classification

Table 1: Top 20 words (after stemming) from four of the topics found by DDM on the Reuters dataset with human-annotated labels.

| Economy | Oil | Technology | Agriculture |
|---|---|---|---|
| mln | price | industri | loan |
| dlrs | oil | comput | tax |
| billion | product | product | payment |
| year | pct | develop | dlrs |
| pct | market | busi | corn |
| loss | cut | market | program |
| oper | barrel | unit | billion |
| net | bond | system | propos |
| note | crude | ibm | plant |
| profit | yield | time | usda |
| compani | corp | line | save |
| share | sourc | sale | bill |
| includ | opec | high | agricultur |
| cts | output | softwar | committe |
| sale | april | profit | texa |
| tax | industri | problem | support |
| march | cost | intern | mln |
| gain | bpd | announc | farm |
| quarter | demand | technolog | year |
| end | manag | design | canada |

Table 2: Top 20 words (after stemming) from four of the topics found by DDM on the NIPS dataset with human-annotated labels.

| Neuroscience | Speech Recognition | Bayesian Statistics | Kernel Methods |
|---|---|---|---|
| respons | speech | bayesian | kernel |
| cell | recognit | prior | svm |
| activ | signal | log | margin |
| stimulus | hmm | posterior | spars |
| synapt | filter | regress | machin |
| synaps | speaker | likelihood | svms |
| cortic | acoust | nois | basi |
| cortex | word | varianc | program |
| stimuli | context | kernel | coeffici |
| trial | markov | regular | laplacian |
| excitabori | frame | densiti | convex |
| inhibitori | mlp | exp | dual |
| simul | recogn | true | page |
| plastic | tempor | mixtur | transduct |
| neurosci | window | risk | dataset |
| dendrit | phonem | hyperparamet | decis |
| inhibit | sequenc | bay | smola |
| membran | utter | evid | denois |
| modul | dynam | fit | choic |
| potenti | frequenc | smooth | sch |

Table 3: Running time of each experiment in hours

| | Offline | Online | | | |
|---|---|---|---|---|---|
| | HDP | Basic MM (Fixed $T$) | DDM | TDM | oHDP |
| Reuters | 3.2 | 0.8 (0.1) | 0.4 | 0.7 | 0.1 |
| NIPS | 29.0 | 0.9 (0.3) | 1.6 | 10.9 | 0.1 |
| Yelp | 25.0 | 2.1 (0.6) | 1.0 | 2.7 | 0.5 |

Table 4: Comparison of AUC scores on Reuters-21578 corpus with word frequencies, DDM, and TDM features

| Topic \ Feature | Words | DDM | TDM |
|---|---|---|---|
| earn | 0.947 | **0.976** | 0.968 |
| acq | **0.945** | 0.895 | 0.885 |
| money-fx | **0.930** | 0.922 | 0.919 |
| crude | 0.958 | **0.985** | 0.979 |
| grain | 0.962 | 0.973 | **0.976** |
| trade | 0.894 | **0.957** | **0.957** |

Table 5: Comparison of AUC scores on Yelp corpus with word frequencies, DDM, and TDM features

| Category \ Feature | Words | DDM | TDM |
|---|---|---|---|
| Coffee & Tea | **0.919** | 0.845 | 0.874 |
| Restaurants | **0.885** | 0.827 | 0.832 |
| Grocery | **0.945** | 0.916 | 0.910 |
| Ice Cream & Frozen Yogurt | **0.948** | 0.848 | 0.919 |

algorithm used was LinearSVC in scikit-learn 0.14.1 [1] with the `class_weight` option set to `'auto'`, which gives each class a weight that is inversely proportional to its frequency in the training set.

Table 4 shows the area under the receiver operator curve (AUC) score for the two models as well as for an SVM that uses word frequencies as features. The result indicates that both models were able to capture the underlying topic distributions as they induced feature sets that had comparative performance with word features despite reducing the feature space from 7,720 dimensions to 39 and 36.

For the Yelp set, we tested classification on the four most common business categories that co-occur with Food. Table 5 shows the AUC scores for using the learned $\Theta$ parameters as features compared to using word frequency counts. In this case, the DDM and TDM features did not do as well as the bag-of-word features. However, the scores were still quite high despite reducing the feature space dimensions by over 99.8% (from 5,640 to 10) for DDM and 99.4% (from 5,640 to 32) for TDM, suggesting that the models did in fact capture semantically meaningful clusters.

## References

[1] F. Pedregosa, G. Varoquaux, A. Gramfort, V. Michel, B. Thirion, O. Grisel, M. Blondel, P. Prettenhofer, R. Weiss, V. Dubourg, J. Vanderplas, A. Passos, D. Cournapeau, M. Brucher, M. Perrot, and E. Duchesnay. Scikit-learn: Machine learning in Python. *Journal of Machine Learning Research*, 12:2825–2830, 2011.