[Reviews · NeurIPS 2016]

Reviewer 1

Summary

The paper presents a new alternative to HDPs for non-parametric Bayesian methods. There is an explicit prior on the number of components. A sequential moment-matching algorithm provides efficient inference.

Qualitative Assessment

I like this approach. Better alternatives to HDPs would be a good contribution. The results are not stunning and could be presented better, but this seems like a very creative solution to the right problem. I'd like to see people talking about it. Ultimately I don't have a lot to say about it. I'd like to try it out, which I think is a good sign. Miller and Harrison have critiqued the inconsistency of the HDP, and I believe at one point suggested an explicit mixture of discrete models as an alternative, but I can't find that reference.

Confidence in this Review

3-Expert (read the paper in detail, know the area, quite certain of my opinion)


Reviewer 2

Summary

The paper combines two ideas: a moment matching algorithm, and a model of "number of topics". Experiments are done on synthetic data and 3 data sets. Experimental method seems robust, but the results do look strange in some cases, and more benchmarks should be used. Algorithms are good and certainly BMM looks pretty good. Some novelty in the models.

Qualitative Assessment

One poorly considered claims in the abstract, though they don't impact the research, so easily fixed and is minor: HDP "is difficult to incorporate prior knowledge since the distri- bution over topics is implicit" HDP has a very well understood prior, and one could call it explicit. Not sure explicit/implicit is a useful distinction as certainly. The number of topics grows logarithmically with the amount of data, as the authors note. Similarly for HPYP. However, HDP is does not say the number of topics is *finite*, which your model does. Arguably, a Bayesian *may* want either model as a prior. Yours is perfectly fine, but you should discuss more. Otherwise, intro is good. e.g., Yes there is a lot of complexity in the HDP implementations, and that is not even starting on the focused topic model (using an IBP) and other more exotic ones. Main problem, however, are that the results look strange. In a lot of experiments I see, Gibbs HDP performs much better than standard LDA, and sometimes its better than oHDP, whereas you get them being very different. Perhaps you need to use Teh's original code which is fast and will run on all your data (I've run it on larger data sets). Or, alternatively, use Mallet's symmetric-asymmetric LDA option which is truncated HDP and performs very well. Given that, in your experiments, Fig 4a, MM substantially beats HDP and oHDP, and we know HDP, when implemented right, substantially beats Gibbs and variational LDA, it either means your HDP and oHDP are wrong, or MM is absolutely amazing. Would have been nice to have seen a standard LDA version (variational) corresponding to your "all T BMM" runs. Also, I'd have thought one probably just needs a logarithmic number of runs in K rather than all T to approximately locate the peak. This way we can assess the quality of BMM versus standard methods. I think this is important and I'd have liked to have seen the results. I would not be surprised if BMM is much better. Novelty is in DDM and TDM and adaptation of BMM to the models. A lot of capitals missing in references. Promising work but poor experimental evaluation. Comments on author comments: > We did not use Teh’s implementation, but we obtained the HDP results using > Wang and Blei's HDP implementation, which is very good and well tested, > so we do not think that is a problem. I disagree. Try the Mallet implementation. > In the experiments, “basic MM” refers to standard LDA where the > # of topics is fixed and parameters are estimated by MM. You have no testing whatever of "basic MM" against an old school standard LDA such as Gibbs or variational, this is needed.

Confidence in this Review

3-Expert (read the paper in detail, know the area, quite certain of my opinion)


Reviewer 3

Summary

The paper aims to propose a simpler model than HDP which will explicitly control the number of topics. The paper is confusing in many parts and lacks minimum essential description to validate the contribution.

Qualitative Assessment

Although the argument around simplicity appears multiple times in the paper but it is kept vague. I assume it means less number of random variables in the joint distribution of the model. However, line 19 indicates me that HDP is complex due to difficulty in understanding its theoretical structure. Then it implies that the aim is to extend LDA generative process by sampling the number of topics. The strategy of the method is to compute likelihood by varying number of topics from 1 to K and pick the one with maximum likelihood. This in some sense does the standard process of selecting number of topics using cross-validation i.e. learn a model with a value for number of topics and then check likelihood on some held-out data and pick the one with maximum likelihood. Here the validation and training sets are same. But due to prior over T, the algorithm can provide some initial bias. The concept is interesting, but initial paragraphs could create confusion. Some examples below. (i) line 5-6, "it is a complex model and it is difficult to incorporate prior knowledge since the distribution over topics is implicit." -- next lines indicate that authors wanted to say the number of topics is not explicitly controlled in HDP, but these lines make it confusing. (ii) "Uniform process may also be used to avoid priors with rich getting richer property" (line 26-27) -- this is correct but is misleading as the context is topic modeling and uniform process does not apply for grouped datasets. The mechanism to share support among multiple uniform processes is still unknown. Such statements should be carefully made. In Eq. 1, the challenge is to compute c_n the normalization factor. In sec 4.1 variational approximation has been introduced, then corresponding sequential update equation to Eq. 1 should be made clear. When P_n is made as prior for (n+1)th sample as mentioned in line 147, then there is some difference with exact posterior which is not discussed here. In this context it is relevant to discuss Online Learning for Latent Dirichlet Allocation, Hoffman et al, NIPS 2010 and Online variational inference for the hierarchical Dirichlet process, Wang et al, AISTATS 2011. Through moment matching the paper could possibly propose alternatives but the essential details have not been discussed here. In Algorithm 1, value of \gamma depends on \alpha and \beta which in turn depend on \gamma,it is not clear how the values are computed for each sample. Currently it seems there is no iteration and they are initialized only once at the beginning that means truncation level for document n depends on other variables till document n-1. Then it confuses me how we are maximizing likelihood at each step. This could be improved by discussing sequential update considering variational approximation and moment matching together. In that sense lines 153-154 is misleading. As this paper is about online method based on moment matching then the content lacks required details. Authors should also note that the number of topics here is controlled by the choice of prior over T or initial value of \gamma, which is regarded as explicit control I think. It is important to discuss that similar behaviour is observed in HDP through the scale variable of DP. That controls the growth in number of topics. If the paper is about explicitness in control over number of topics, then this part requires some discussion.

Confidence in this Review

2-Confident (read it all; understood it all reasonably well)


Reviewer 4

Summary

This paper presents two new models that extend LDA such that both the number of topics and other model parameters can be learned. The new models are shown to outperform non-parametric models such as HDPs.

Qualitative Assessment

The extension to LDA, as shown in figure 1, is quite straightforward, which is good. But the assumption that K is known makes the models a bit weak. In a sense, you convert the problem of determining the number of topics T to a problem of determining the max number of topics K. I am not convinced that the latter can solved more effectively, esp. considering that the computation cost of the new models depends on K. The experiments are a bit weak. It would have been more convincing if the evaluation can be performed on e2e tasks where topic models play an important role, such as language modeling, doc retrieval.

Confidence in this Review

2-Confident (read it all; understood it all reasonably well)


Reviewer 5

Summary

The paper addresses an important problem in topic modeling: selection of number of topics. Despite the popularity of topic modeling there is still no universal solution for this problem. Two methods for estimation of topics number in topic modeling are proposed in this paper. Both methods assume it is possible to set in advance a maximum number of topics (we can always set this number to data dimensionality). In the first method the discrete distribution for the actual number of topics is assumed, while the second method considers all possible numbers of topics that are not greater than the fixed maximum one. For inference the authors propose Bayesian online moment matching algorithm between a true posterior of hidden parameters, that are the word-topic and topic-document distributions and the number of topics, and an approximated one. The approximated distribution is factorised in order to separate the parameters. The inference algorithm is online, i.e. process data sequentially.

Qualitative Assessment

The paper proposes an interesting approach for selection of topics number. I would suggest to change the motivation of the proposed approach (in comparison to the well-known HDP) in the introduction and the abstract. The arguments "it [HDP] is a complex model" or "Existing inference algorithms for HDPs ... are also fairly complex" do not look convincing. The proposed approach and the HDP have conceptually different ideas behind. In the proposed approach there is a direct prior on the number of topics, whereas in the HDP we model growth of the number of topics w.r.t. the data dimensionality. This idea is mentioned in the paper (sentence "Furthermore, the implicit distribution ... " line 22) but the complexity is more emphasised. It would be better to add more background in section 4: how moment matching is connected to approximation of the posterior distribution and where Assumed Density Filtering mentioned in section 2 comes. The expected value of the Kronecker delta requires some clarification. In the inference part it would be better to provide more details about the derivation of the resulting formula (10)-(15). The reason of the claim at the end of the section "As a result, the DDM has a natural bias towards smaller number of topics" is unclear. Section 5 provides thorough experiments on synthetic and three real datasets. There are ways for improvements. The results on synthetic data are not very impressive. Not all hyperparameters settings are mentioned, i.e. alpha, beta, gamma (for the HDP). It is also unclear whether the methods are set with closest settings as possible. The number of recovered topics for the HDP is only mentioned for Yelp data. It is unclear why HDP and oDHP are not on figure 4b. The paper is well and clearly written. Although the description of the triangular Dirichlet model requires several readings. It would be better, for example, to specify what are the first, second and third dimensionality of Theta before referred to them by order. The last sentence in the conclusions referred to the results that are not in the main paper - this looks strange. Minor things: 1. In the footnote on the page 3, should it be {d_n}_{n=1}^N (N instead of D)? 2. Line 104: "... we can SEE that after SEEING ... " 3. Check style format for page title 4. Citations for a stemmer and stopwords would be good 5. There is a duplicate in references ([4] and [5])

Confidence in this Review

2-Confident (read it all; understood it all reasonably well)